# Evaluation and screening of technology start-ups based on PCA and GA-BPNN

Jiaxin Li[1]☉, Mingming Meng[1]☉, Xin Liu[1], Yanjie Lv[2], Jian Yu[1,3]*

**1** School of Economics & Management, Beijing Forestry University, Beijing, P.R. China, **2** School of Information Science and Technology, Beijing Forestry University, Beijing, P.R. China, **3** Institute of New Rural Development, Guizhou University, Guiyang, Guizhou, P.R. China

☉ These authors contributed equally to this work.

\* yja09@tsinghua.org.cn

**Data Availability Statement:** All relevant data are within the paper and its Supporting Information files.

**Funding:** This work was supported by the National Key Research and Development Program of the

## Abstract

### Purpose

Due to the existence of information opacity, there is a common problem of adverse selection in the process of screening alternative technology start-ups (TSs) and determining investment targets by venture capital institutions, which does not reveal the true value of enterprises and makes the market inefficient. The aim of this paper is to design an evaluation and screening system help venture capital institutions to select the qualified TSs as their investment objective.

### Design

A research framework of four dimensions that include conception, technical innovation, business model and team structure, was built based on previous studies. Based on the research framework, 15 second-level indicators and 33 third-level indicators were extracted with literature research method. This paper proposes an evaluation model with back propagation neural network (BPNN) optimized by genetic algorithm (GA) to improve the rate of selecting and investing in qualified start-ups.

### Findings

The results show that the evaluation accuracy of the evaluation model for qualified and unqualified enterprises can reach 80.33% and 93.67% respectively, which has verified the effectiveness of the model and algorithm.

### Originality/Value

This paper established an effective evaluation system based on PCA and GA-BPNN to help venture capital institutions preliminarily screen potential technology start-ups, which provides the possibility for venture capital institutions to greatly reduce the screening time and cost, improve the screening efficiency of TSs, and scientifically assess the risk of investee projects or investee enterprises to obtain sustainable and stable excess profits.

Ministry of Science and Technology of China (No. 2017YFB1402000). The funders had no role in study design, data collection and analysis, decision to publish, or preparation of the manuscript.

**Competing interests:** The authors have declared that no competing interests exist.

## 1. Introduction

In today's world, science and technology play a vital role in economic and social development. The development trend of TSs has become the focus that the governments and scholars have been paying attention to. Governments and scholars are increasingly recognizing that technology entrepreneurship (TE) is an important engine for developing emerging industries and promoting high-quality economic development. The vigorous development of TSs helps to restructure the value chain of the industry, being the key in the future in the process of accelerating the evolution under the international economic pattern. In September 2014, the then Premier Li Keqiang announced at the Summer Davos Forum in Tianjin that "Widespread Entrepreneurship and Innovation" would be a new engine for China to upgrade the economy and make it more efficient. Since then, Chinese start-ups, especially high-tech ones, have made great progress. Under the guidance of innovation and entrepreneurship policies, TSs in China are forming a trend, which is the source of power to improve the overall quality of the economy and international competitiveness.

As the engine of economic growth, venture capital can provide an effective source of capital for the transformation of scientific and technological achievements (STAs) for start-ups. However, the "bottleneck" of financing seriously restricts the vigorous development of start-ups, which urgently need the support of venture capital. It is a common phenomenon that start-ups, especially TSs, have a low success rate. The production of scientific and technological innovation achievements is highly difficult and risky, which requires long-term R&D investment. The incubated TSs, with higher risk of failure, are in more urgent need of getting venture capital to support the transformation of STAs, and then push them to the market quickly to gain competitive advantages. However, when faced with a large number of TSs, venture capital institutions still face difficulties in assessing and screening them quickly and accurately. To minimize the problem of adverse selection that is prevalent when investing in start-ups, it is crucial for venture capital institutions to build an algorithm to reduce the screening cost of investment targets. This difficulty is reflected in the fact that assessment and selection is highly subjective, and it is also possible that adverse selection problems may arise when identifying investment targets. How to evaluate and screen scientifically, effectively and accurately of TSs has become an important issue for venture capital institutions.

Designing a good system of evaluating and screening TSs has become one of the key aspects of the investment process for venture capital institutions. The establishment of an evaluation and screening system can provide a series of alternative potential partners for venture capital institutions, thus helping venture capitalists in a disadvantaged information position to make strategic investment decisions, reduce information opacity in the process of achieving cooperation, and overcome the problem of adverse selection. Therefore, this paper takes TSs as the research object and constructs an evaluation model based on PCA and GA-BPNN. This model is important for venture capital institutions to evaluate the value of TSs and improve the correctness of investment decisions, as well as to select the best investment target effectively. The remainder of the paper is structured as follows: Section 2 reviews the existing literature and constructs the index system. Section 3 performs principal component analysis. Section 4 investigates the BPNN model (BPNNM) and derives preliminary simulation results. In section 5, the GA is selected to optimize the BPNNM to produce simulation results with higher evaluation accuracy. Finally, section 6 presents the conclusions, and practical implications.

## 2. Literature review and index system construction

### 2.1 Review of related studies

TE is a special form of entrepreneurship. In the process of economic development, TE can promote technological innovation, adjust the industrial structure, drive the development of

entrepreneurial venture capital, and guide the economic demand and supply-side structural reform. The era of production capacity upgrading and economic structure optimization starting from TSs has arrived, and TSs have made great contributions to job creation and economic development [1].

TE not only has the characteristics of general entrepreneurship, but also is unique. Nacu and Avasikai [2] found that all the definitions of TE are slightly different, but they all have one thing in common, that is, the spirit of adventure, and the new business philosophy in the field of technology. TE is generated based on academic entrepreneurship, technological entrepreneurship, university-derived enterprise-level scientist entrepreneurship and other concepts, all of which have one thing in common—technology-driven [3]. Whereas, the introduction of various supporting policies and measures has promoted the development of TE, but also spawned the phenomenon of "pseudo-TE". In addition, there are some phenomena occurring frequently, such as serious homogeneity of innovation achievements, plagiarism and running away of makers [4]. This also reflects that it is particularly important to improve the efficiency and accuracy of "identifying" and "screening" TSs by "incubators". However, at present, the academic research on TSs is more inclined to the study of single-factor or multi-factor interrelationship, and there are few research results on the value assessment and evaluation of TSs.

For instance, Santisteban & Mauricio [5] conducted a systematic literature review of key success factors for TSs, and identified them through empirical research. Santisteban et al. [1] identified key success factors for TSs based on the analysis of the information system theory by using human, social, and organizational behavior theory. Kim et al. [6] investigated experts of TSs by using the Analytic Hierarchy Process (AHP) and found that commercialization of ideas, continuous investment and market-oriented opportunity conversion are the most critical success factors for start-ups, and further found that continuous investment was the most important factor, followed by commercialization of ideas. Lin et al. [7] proposed a new hesitation-fuzzy language decision method to solve the evaluation problem of TSs.

Small businesses, such as TSs, are frequently characterized by information opacity. And the opacity results in a possible adverse selection for venture capital institutions on investment targets [8]. Venture capital institutions need to overcome the problem of adverse selection in the process of screening TSs and determining investment targets. It is important to understand that a large number of economic, social, technological, ecological, environmental and other indicators define the effectiveness of the investment process [9]. According to some of the indicators, some startups are suitable for being selected and invested, while they are not according to other indicators. Certain characteristics of the enterprises (e.g., their young age, small capitalization, and simplified accountancy requirements) make it hard for outside investors to verify their financial situation and investment decisions [8]. In order to seek better and more accurate evaluation methods and strengthen the construction of credit enhancement mechanism in venture capital, it is necessary to establish a perfect evaluation and screening system and increase the realization rate of venture capital.

From the current studies, although some scholars have summarized the factors influencing the development of TSs [1,10], there is a lack of systematic and perfect screening and assessment system studies. At present, the existing studies have the following two problems. First and foremost, TSs are greatly influenced by environmental policies, and the update and iteration speed of these policies are fast, so, the earlier research results lack a scientific and effective system for evaluating and screening TSs and are not applicable to the current TSs.

The field of machine learning is now expanding rapidly at the intersection of computer science and statistics, and forms the core of artificial intelligence and data science [11]. Recent advancements in machine learning have been driven by the development of new learning algorithms. For example, Zheng et al. [12] proposed an adaptive memetic differential evolution-

back propagation-fuzzy neural network. Additionally, Xue et al. [13] combined differential evolution with Adam to create ensemble of differential evolution and Adam, which integrates a modern version of the differential evolution algorithm with Adam.

Machine learning approaches are widely adopted in various fields including science, technology, and industry. These approaches contribute to evidence-based decision-making in healthcare, biomedicine, manufacturing, education, financial modeling, data governance, policing, and marketing [14,15]. In specific application of machine learning, Du et al. [16] proposed a hybrid model that combines long short-term memory with discrete wavelet transform and principal component analysis (PCA) pre-processing techniques for water demand forecasting. They aimed to improve the accuracy of the forecasting process. In another study, Ozcalici and Bumin [17] used GA to optimize the parameters of filtering rules and stock selection for enhanced trading. They integrated artificial neural networks into their approach to achieve better trading performance. Furthermore, Zhu et al. [18] developed a hybrid variational mode decomposition and stacked gated recursive unit model that consisted of a prediction module and an environment module to predict trading strategies for a target stock's closing price. These studies highlight the diverse application of machine learning in different domains, from water demand forecasting to stock trading, and the various techniques used to enhance its performance.

The existing studies lack more in-depth simulation operations. Whether it is for evaluating and screening TSs or general start-ups, scholars mostly used AHP to evaluate them, subjective factors account for a large proportion, and it is difficult to guarantee the objectivity of the evaluation. Therefore, it is of great significance and value to re-establish an evaluation system of TSs in the current environmental background. Combined with the latest progress and specific application of machine learning methods, this study chooses to use the BPNNM and GA to evaluate the TSs (project) from a new perspective of research ideas and methods.

## 2.2 Construction of the index system

**2.2.1 Index arrangement.** In order to build a suitable index system for evaluating and screening TSs, adopts the literature research method was adopted this paper to select the first-level indicators by collecting the existing research results as far as possible to complete the indicators sorting, and then further screening and summarizing indicators. Therefore, this paper refers to many literatures related to the evaluation and selection of start-ups, and classifies the existing index system into two types: general start-ups and TSs. The indicators are summarized in Table 1

**2.2.2 Selection of indicators.** Combined with the actual situation of TSs, based on the existing literatures analysis, and combined with the characteristics of TSs, to select the first-level indicators.

1. We sort out the existing assessment and evaluation index system of (technology) start-ups/ projects respectively according to the two categories of technology start-ups/ projects and general start-ups/ projects. To a certain extent, the indicators of technology start-ups/ projects are derived and optimized based on the indicators of general start-ups / projects. Therefore, this paper refers to the index system of general start-ups/ projects. Meanwhile, although the index system of technology start-ups/ projects has a high reference value, it is necessary to establish an evaluation system suitable for the current TSs due to the characteristics of rapid technology iteration and high risk.

2. The evaluation indicators of start-ups tend to be consistent in the selection of environment, entrepreneurial team, technology, product and economic indicators. Most of the research

**Table 1. (Technology) start-ups/ project assessment and evaluation index system.**

| Type of enterprises | The first-level indicators | | Source of indicators |
|---|---|---|---|
| General start-ups/ projects | Macro environment, Micro environment, Internal environment, Enterprise innovation characteristics | | [19] |
| | Innovation, Entrepreneurship, Resources, Capacity, Intellectual capital, Sustainable development, Content management, Information advantage | | [20] |
| | Entrepreneurial orientation, New product development, Technological innovation | | [21] |
| Technology start-ups/ projects | Culture of innovation and entrepreneurship, Phased financing, Support from business incubators, Customer satisfaction, Innovation and entrepreneurship ecosystem, Dynamic capabilities | | [1] |
| | Dimensions of environmental dynamism | Innovation, Changing tastes, Technological advances | [10] |
| | Uncertainty in the operating environment | Socio-economic conditions, Competitive hostility | |
| | Dimensions of business environment | Heterogeneity, Interaction | |
| | Organization | Product innovation, Aggregation | [5] |
| | Personal | Industry experience in setting up a team, Entrepreneurial characteristics | |
| | External | Venture capital, Technology policy | |

factors for TSs are the research on policy support for these enterprises, scientific and technological innovation ability, various characteristics of entrepreneurs (such as cognitive style, personality traits, etc.), business model, resources, financing, entrepreneurial team, and other aspects.

From above analysis, it can be basically summarized into the following five dimensions: conceptual architecture, technological innovation, business model, team structure, and operation status to evaluate a business. However, in order to evaluate and select a qualified TS located in the start-up period, the evaluation system with five dimensions should be changed.

While some of Chinese high-tech enterprises and network ones are still in the state of no profit or micro profit before or at present. Meanwhile, for the new company, the lack of historical data means that part of the business information is in a blank state [22]. Therefore, the business status dimension is eliminated. Thereby, four dimensions of conceptual architecture, scientific and technological innovation, business model and team structure are eventually selected as the first-level evaluation indicators of the evaluation system of TSs in this paper.

In addition, the selection process of second-level and third-level indicators is the same as above, and the research results of various dimensions are sorted out via four first-level indicators. In combination with the identification conditions of TSs, the actual situation of China's TSs, the identification conditions of high-tech enterprises, and the data investigation reports of TSs in various regions or industrial parks, several key index factors are selected to initially obtain the evaluation system of TSs.

We employed the Delphi technique to further determine the indicators and their weights. Specifically, we selected six professors from universities and nine managers from related companies. All respondents were over 30 years old. Five professors from universities were all doctoral supervisors, and their research directions were all business management. The nine managers had more than five years of experience in technology-based small and medium-sized enterprises. We solicited opinions from the aforementioned experts through online meetings, distributed survey outlines and background resources in advance, and answered

some experts' questions. We summarized the experts' initial discussion opinions on the indicator system and fed back the results to the experts for the second revision. After four rounds of anonymous discussion, the indicator system was finally determined, and the experts' opinions were unified. In the subsequent fifth and sixth rounds of discussion, we asked the experts to provide importance evaluations for the three-level indicators of the determined indicator system, thus further determining the corresponding scores for each indicator. The final evaluation system and index scores for TSs include 4 first-level indicators, 15 second-level indicators, and 33 third-level indicators. As shown in Table 2.1

**Table 2.1. Evaluation index system of TSs.**

| First-level indicators | Second-level indicators | Third-level indicators | Code | References |
|---|---|---|---|---|
| Conceptual architecture | Industrial demand | Guiding category of industrial Structure Adjustment (2 points) | $X_1$ | [5,20] |
| | | Green Industry Guidance Catalogue (2019 edition) (2 points) | $X_2$ | |
| | | Catalogue of Key Intellectual Property Support Industries (2018 edition) (2 points) | $X_3$ | |
| | Technological innovation | Key support category of the Ministry of Science and Technology (6 points) | $X_4$ | [23] |
| | Strategic support | The Made in China 2025 Green Paper on Technological Innovation in Key Areas (3 points) | $X_5$ | [5] |
| | | Guidance Catalogue of Key Products and Services for Strategic Emerging Industries (2016 edition) (3 points) | $X_6$ | |
| | Concept value | National or provincial awards or relevant national association awards or achievements have been published by authoritative journals in relevant fields at home and abroad or recognized by well-known leading enterprises in the industry (6 points) | $X_7$ | [24] |
| | Capital market | Whether the foreign financing has been successful (3 points) | $X_8$ | [1,5,6,20] |
| | | Whether the financing valuation exceeds 100 million yuan (3 points) | $X_9$ | |
| Technical innovation | Team technology research and development strength | Whether the funds used for the research and development of new and high technology and its products should account for more than 3% of the total annual income of the enterprise (2 points) | $X_{10}$ | [25–28] |
| | | Whether the sum of technological income and output value of high-tech products of a high-tech enterprise should account for more than 50% of the total income of the enterprise in the current year (2 points) | $X_{11}$ | |
| | Product and technical strength | Whether the invention patent or new drug approval related to the main business or whether the computer software related enterprises have software copyright (3 points) | $X_{12}$ | [6,20,24,26] |
| | | Whether to lead the scientific research projects of the National Natural Science Foundation of China or ministries and commissions (3 points) | $X_{13}$ | |
| | | Whether to sponsor or participate in the formulation of national standards or industrial standards (3 points) | $X_{14}$ | |
| | | Whether to win the national Science and Technology Award (2 points), the industry authority award (1 point) | $X_{15}$ | |
| Business model | Own business maturity | Whether there are comparable domestic and foreign listed companies in the industry (2 points) | $X_{16}$ | [5,25] |
| | Industry chain grade of maturity | Whether there are comparable domestic and foreign listed companies in the upstream industry (2 points) | $X_{17}$ | [5,29] |
| | | Whether there are comparable domestic and foreign listed companies in the downstream industries of the industry (2 points) | $X_{18}$ | |
| | Industry chain relation | Whether the closely cooperated upstream suppliers have domestic and foreign listed companies or world top 500 or Top 500 in China (3 points) | $X_{19}$ | [29] |
| | | Whether close cooperation with downstream suppliers have domestic and foreign listed companies or world top 500 or Top 500 in China (3 points) | $X_{20}$ | |
| | Ecological relationship | Whether they have public cooperation records with ministries or provincial government agencies (including provincial capitals, cities separately listed in the state plan, special economic zones) or the world top 500 enterprises (3 points) | $X_{21}$ | [1,20] |
| | | Whether it have public cooperation records with municipal government agencies and Chinese top 500 enterprises (2 points) | $X_{22}$ | |
| | | Whether to participate in the national industry or industry strategic alliance and industry-related self-discipline association organizations (2 points) | $X_{23}$ | |

(*Continued*)

**Table 2.1.** (Continued)

| First-level indicators | Second-level indicators | Third-level indicators | Code | References |
|---|---|---|---|---|
| Team structure | Team strength | Academicians of the Chinese Academy of Sciences or 863 Chief or 973 Chief or Changjiang Scholars or Outstanding Young Foundation of China or experimental leaders of key disciplines (2 points) | $X_{24}$ | [5] |
| | | Professor or Associate Professor or Local Talent Program (1 point) | $X_{25}$ | |
| | | Whether to hold the decision-making or management position of an important academic organization or alliance in the industry (1 point) | $X_{26}$ | |
| | Team quality | Whether the scientific and technological personnel with a bachelor degree or above account for more than 30% of the total number of enterprise employees (2 points) | $X_{27}$ | [5,27] |
| | | Whether the scientific and technological personnel engaged in the research and development of high and high-tech products account for more than 10% of the total number of employees or engaged in the production of high-tech products or service labor-intensive high-tech enterprises, whether the scientific and technological personnel with college degree or above account for more than 20% of the total number of employees (2 points) | $X_{28}$ | |
| | | Whether the main team members have some industry and upstream and downstream working experience (1 point) | $X_{29}$ | |
| | | Whether the main team members have some industry and upstream and downstream research experience (2 points) | $X_{30}$ | |
| | | Whether there are experienced personnel in the main team members (2 points) | $X_{31}$ | |
| | Team stability | The proportion of directors, senior management or core technical personnel leaving in the past 2 years is less than 33% (2 points) | $X_{32}$ | [17,27] |
| | Shareholder strength | Whether the shareholders have a well-known or large organization or individual (3 points) | $X_{33}$ | [5] |

Prior to initiating the formal research of this paper, specifically prior to distributing the survey questionnaire, eight experts were requested to evaluate the validity of the questionnaire's structure, content, and overall design. Based on the evaluation results from these eight experts (see Table 2.2), they provided positive feedback on the questionnaire structure, content, and overall design. Therefore, the survey questionnaire designed in this paper has a high level of validity.

**2.2.3 Description of the index system.** The main contents are as follows:

1. Conceptual architecture, including industrial demand, technological innovation, strategic support, conceptual value, and capital markets. Whether the main products and technologies of TSs are in line with the various categories of national support and the scope of relevant guidance catalogue, which is an important guarantee for determining whether they can continue to operate and enjoy the relevant national policies and treatment in the future. Starting from the concept of entrepreneurship, consider its value, demand, strategic support and whether they belong to scientific and technological innovation, and consider the

**Table 2.2. Result of experts' evaluation (N = 8).**

| | | Fairly reasonable | Basically sound | Ordinary | Somewhat unreasonable | Completely unreasonable |
|---|---|---|---|---|---|---|
| Structure | Number of choices | 3 | 5 | 0 | 0 | 0 |
| | Percentage | 38% | 63% | 0% | 0% | 0% |
| Content | Number of choices | 2 | 6 | 0 | 0 | 0 |
| | Percentage | 25% | 75% | 0% | 0% | 0% |
| Overall design | Number of choices | 2 | 6 | 0 | 0 | 0 |
| | Percentage | 25% | 75% | 0% | 0% | 0% |

capital factors, and show them from the financing situation. Therefore, it is the subdivision index of the conceptual architecture of TSs.

2. Technological innovation, including team technology research and development strength, product, and technical strength. Technological innovation has changed the status of the various pillar industries, caused the replacement of leading industries, transformed traditional industries, and accelerated the decline of old industries. Technological innovation can also make some industries have explosive growth, thus become the new pillar industries. The core of TSs is technological innovation. The analysis of technological innovation for TSs will be measured from the two aspects of technology and product, so this paper is divided into technology research and development strength and product and technology strength. The sources of technology research and development indicators are mainly the identification conditions of TSs and high-tech enterprises, while the aspect of product and technology is more consider the patent or software copyright, the undertaking of scientific research projects, and the obtaining of authoritative awards.

3. Business model, including its own business maturity, industrial chain maturity, industrial chain relationship and ecological relationship. In order to better evaluate the overall strength of TSs, this paper refers to the research results of Wu & Shi [30] and Santisteban & Mauricio [5] when selecting business model indicators, and does not include traditional enterprise profit model, key resource capability and other indicators into the index system. Instead, it pays more attention to the industrial chain, their own business, and the ecological relationship of TSs, and measures the development of their industry, the development of upstream and downstream industries, the quality of suppliers and customers they cooperate with, and whether they have the experience of government-enterprise cooperation.

4. Team structure, including team strength, team quality, team stability, and shareholder strength. Industry selection, development mode, system and team are the three major elements of entrepreneurial success. The scientific and technological independent innovation ability of enterprises cannot be separated from scientific research organizations and scientific and technological talent teams. The team is not only the beginning of an enterprise but also the guarantee of its development. The strength of a team should be investigated from whether there are leading experts, professors, and associate professors and the like, to evaluate the availability of its scientific research projects. Part of the indicators of team quality come from the identification conditions of TSs. Meanwhile, the experience and background of team members is investigated. Furthermore, the condition of the turnover ratio of members in the issuance conditions of the Science and Technology Innovation Board is taken as an index to measure the team stability, and the strength of shareholders is also taken as a major evaluation index.

### 2.3 Data source

In order to ensure the authenticity and reliability of the data obtained, the sample enterprises selected in this paper are all listed enterprises. However, since all TSs are small and medium-sized enterprises, this paper only selects listed enterprises from National Equities Exchange and Quotations of China (commonly known as "new OTC market"). The selection criteria are as follows:

1. The age of the listed TSs (LTSs) on the new OTC market was no more than 5 years after its establishment (excluding public service and professional consultation enterprises);

2. Their average annual asset growth rate, operating revenue growth rate and employee number growth rate shall be no less than 15%, 15% and 5%, respectively, since they were listed;

3. Since listed, their worst annual growth rate of assets, worst annual growth rate of operating revenue and annual growth number of employees shall be no less than -5%, -5% and -5%, respectively.

According to these above criteria, 240 LTSs are initially selected, they are involved in high-end equipment manufacturing industry, information technology industry, related service industry, energy conservation and environmental protection industry, new materials industry, information technology industry, medical industry, biological industry, and new energy industry.

In accordance with the principle of rationality and scientificity, the preliminary sample data collected are sorted out and screened. From the perspective of the financial status and scale growth of enterprises, we divide enterprises into qualified enterprises and unqualified enterprises. Qualified enterprises represent enterprises with normal financial status and growth, while unqualified enterprises represent enterprises that are lagging behind or stagnating. These LTSs are classified into two groups that are qualified and unqualified start-ups, according to whether the number of employees, total assets and total income have maintained a growth rate of more than 10% in 2017 and 2018 compared with those in the previous year. Finally, data of 200 sample start-ups are obtained, and among these samples, 100 start-ups are qualified enterprises, and 100 unqualified.

## 3. Principal component analysis (PCA)

PCA, is a common data analysis method [31,32]. By using linear algebra method to reduce the dimension of multiple variables of the original data, based on which fewer independent variables are obtained, and these variables obtained are the principal components [33]. The information represented by each principal component will not overlap and can describe the whole data set in a relatively comprehensive way [34]. As can be seen from Table 2, the number of third-level indicators are high, and it is difficult to determine whether every indicator is independent or multicollinear with other indicators. In order to make the results obtained more scientific and accurate for this evaluation system, PCA method is used for the final sample data to extract the principal components and eliminate the multicollinearity between indicators. In order to make the results obtained more scientific and accurate for this evaluation system, PCA method is used for the final sample data to extract the principal components, filter the input variables of the model from 33 to 22, and eliminate the multicollinearity between indicators.

### 3.1 Conditions of the principal component analysis

SPSS 26.0 software is applied in this paper to input conduct the statistical analysis. The KMO (Kaiser-Meyer-Olkin) and Bartlett's test results are shown in Table 3.

Table 3. KMO and bartlett's test.

| Kaiser-Meyer-Olkin measure of sampling adequacy. | | 0.527 |
|---|---|---|
| Bartlett's test of sphericity | Approx. Chi-Square | 1 097.647 |
| | df | 528 |
| | Sig. | 0 |

Table 4. Eigenvalues, variance contribution rate and cumulative contribution of variance.

| Principal component | Eigenvalue | Variance contribution rate (%) | Cumulative contribution rate of variance (%) |
|---|---|---|---|
| $F_1$ | 2.977 | 9.021 | 9.021 |
| $F_2$ | 2.233 | 6.767 | 15.788 |
| $F_3$ | 1.962 | 5.945 | 21.733 |
| $F_4$ | 1.696 | 5.139 | 26.872 |
| $F_5$ | 1.580 | 4.788 | 31.661 |
| $F_6$ | 1.544 | 4.680 | 36.340 |
| $F_7$ | 1.486 | 4.503 | 40.843 |
| $F_8$ | 1.388 | 4.205 | 45.048 |
| $F_9$ | 1.258 | 3.894 | 48.942 |
| $F_{10}$ | 1.227 | 3.718 | 52.660 |
| $F_{11}$ | 1.187 | 3.597 | 56.257 |
| $F_{12}$ | 1.112 | 3.369 | 59.626 |
| $F_{13}$ | 1.076 | 3.262 | 62.888 |
| $F_{14}$ | 0.989 | 2.996 | 65.884 |
| $F_{15}$ | 0.946 | 2.865 | 68.749 |
| $F_{16}$ | 0.923 | 2.797 | 71.545 |
| $F_{17}$ | 0.888 | 2.691 | 74.237 |
| $F_{18}$ | 0.855 | 2.592 | 76.829 |
| $F_{19}$ | 0.746 | 2.260 | 79.089 |
| $F_{20}$ | 0.714 | 2.164 | 81.253 |
| $F_{21}$ | 0.707 | 2.143 | 83.396 |
| $F_{22}$ | 0.676 | 2.047 | 85.443 |

Table 4 clearly shows that the cumulative contribution rate of factors with eigenvalue λ greater than 1 is only 62.888%. Therefore, 22 factors with cumulative contribution rate of variance greater than 85.000% are selected as the main components in this paper.

From Table 3, the KMO value is greater than 0.5, and the probability of significance in bartlett's test of sphericity is much less than 0.05. Thus, the sample data can meet the requirements of the PCA method, indicating that the data can be used for PCA.

### 3.2 Results of the principal component analysis

According to the principle of PCA and research experiences, the factors with the cumulative contribution rate of variance greater than 85% or eigenvalue λ of the software greater than 1 are generally selected as the principal components. The results of the eigenvalue, and cumulative contribution rates of variance of the sample data set are shown in Table 4.

### 3.3 Principal component extraction

The factor load matrix is shown in Table 5. According to the principle of PCA method, the relationship between the original indicator $X_1 X_2 \cdots X_{32} X_{33}$ and the principal component $F_f$ is:

$$F_f = L_{f1}X_1 + L_{f2}X_2 + L_{f3}X_3 + \cdots + L_{f31}X_{31} + L_{f32}X_{32} + L_{f33}X_{33} \tag{1}$$

where, the right subscript, $f$, of $L_{fi}$ indicates the serial number of the principal component, and $i$ indicates the load number of the principal component. All indicator data is then substituted into Formula (1), and a partial display of the results for each principal component ($f$ = 1,2, ..., 22) is provided below. The partial values of $F_f$ obtained are presented in Table 6, and the

**Table 5. Principal component factor load matrix.**

| Influential factors | Ingredients | | | | | | | | | | | | | | | | | | | | | |
|---|---|---|---|---|---|---|---|---|---|---|---|---|---|---|---|---|---|---|---|---|---|---|
| | $F_1$ | $F_2$ | $F_3$ | $F_4$ | $F_5$ | $F_6$ | $F_7$ | $F_8$ | $F_9$ | $F_{10}$ | $F_{11}$ | $F_{12}$ | $F_{13}$ | $F_{14}$ | $F_{15}$ | $F_{16}$ | $F_{17}$ | $F_{18}$ | $F_{19}$ | $F_{20}$ | $F_{21}$ | $F_{22}$ |
| $X_1$ | -0.38 | 0.39 | 0.31 | -0.02 | 0.39 | -0.02 | 0.06 | 0.05 | -0.22 | 0.07 | -0.06 | -0.09 | -0.15 | 0.08 | 0.22 | -0.09 | -0.28 | -0.07 | 0.03 | -0.02 | -0.01 | -0.15 |
| $X_2$ | -0.58 | 0.16 | -0.11 | 0.37 | 0.36 | 0.09 | -0.02 | -0.11 | -0.12 | 0.07 | 0.08 | 0.12 | -0.06 | 0.10 | 0.05 | -0.16 | 0.06 | 0.02 | 0.02 | -0.14 | -0.02 | 0.17 |
| $X_3$ | -0.03 | 0.20 | 0.43 | -0.10 | 0.20 | -0.20 | 0.02 | -0.37 | -0.16 | 0.25 | -0.11 | 0.30 | -0.11 | 0.19 | -0.06 | 0.14 | 0.29 | 0.21 | 0.05 | 0.03 | 0.10 | -0.03 |
| $X_4$ | -0.19 | 0.05 | -0.17 | 0.35 | 0.16 | 0.50 | 0.14 | -0.18 | -0.09 | 0.00 | -0.37 | 0.11 | 0.01 | -0.02 | -0.03 | -0.10 | 0.04 | -0.13 | 0.38 | -0.02 | 0.10 | -0.03 |
| $X_5$ | 0.09 | 0.28 | 0.51 | -0.28 | 0.10 | -0.44 | 0.07 | -0.09 | 0.04 | 0.08 | -0.24 | 0.03 | -0.04 | 0.18 | 0.01 | 0.14 | -0.08 | -0.16 | 0.00 | 0.05 | 0.13 | -0.07 |
| $X_6$ | 0.20 | 0.17 | 0.06 | 0.44 | 0.19 | -0.13 | -0.12 | -0.12 | 0.30 | -0.24 | 0.18 | -0.27 | -0.08 | 0.20 | -0.34 | -0.05 | 0.10 | -0.11 | 0.01 | 0.14 | 0.05 | -0.29 |
| $X_7$ | 0.51 | -0.04 | 0.34 | 0.23 | -0.31 | 0.02 | 0.05 | -0.35 | -0.04 | -0.02 | 0.01 | 0.07 | 0.05 | 0.00 | -0.02 | -0.18 | 0.00 | 0.20 | 0.04 | -0.14 | 0.06 | 0.12 |
| $X_8$ | 0.60 | 0.08 | -0.08 | -0.06 | 0.07 | -0.01 | -0.08 | 0.14 | -0.04 | 0.02 | 0.01 | -0.15 | -0.15 | 0.20 | -0.37 | -0.03 | -0.13 | 0.04 | 0.21 | -0.01 | 0.16 | 0.24 |
| $X_9$ | 0.32 | 0.03 | -0.17 | -0.21 | 0.29 | 0.24 | -0.09 | -0.10 | -0.20 | 0.12 | 0.39 | -0.12 | -0.10 | 0.38 | 0.10 | 0.02 | 0.03 | -0.18 | -0.16 | -0.20 | 0.00 | 0.11 |
| $X_{10}$ | 0.29 | -0.33 | 0.02 | 0.05 | 0.15 | 0.07 | 0.55 | 0.21 | 0.03 | -0.09 | -0.07 | 0.17 | -0.19 | 0.14 | 0.26 | -0.13 | -0.21 | -0.17 | 0.05 | 0.06 | 0.09 | 0.08 |
| $X_{11}$ | 0.14 | -0.46 | 0.12 | 0.08 | 0.15 | -0.15 | 0.53 | 0.16 | 0.09 | -0.14 | 0.01 | 0.04 | 0.00 | -0.06 | -0.01 | 0.01 | 0.28 | -0.11 | -0.23 | 0.01 | 0.32 | 0.06 |
| $X_{12}$ | -0.04 | 0.19 | 0.23 | 0.33 | 0.27 | -0.11 | -0.08 | -0.14 | 0.16 | 0.04 | 0.36 | -0.06 | 0.13 | -0.43 | -0.04 | 0.07 | -0.38 | -0.14 | 0.03 | 0.06 | 0.26 | 0.15 |
| $X_{13}$ | -0.08 | -0.25 | 0.15 | 0.18 | -0.10 | 0.02 | 0.37 | -0.30 | -0.32 | -0.02 | 0.44 | 0.05 | 0.05 | 0.17 | -0.19 | 0.04 | -0.05 | 0.06 | 0.03 | 0.21 | -0.28 | 0.10 |
| $X_{14}$ | 0.03 | 0.03 | 0.03 | 0.04 | 0.41 | 0.10 | 0.28 | 0.04 | 0.04 | 0.05 | -0.40 | -0.14 | 0.50 | 0.09 | -0.34 | -0.03 | -0.07 | 0.13 | -0.18 | -0.04 | -0.15 | 0.12 |
| $X_{15}$ | 0.34 | -0.08 | 0.10 | 0.27 | -0.20 | -0.08 | -0.24 | -0.11 | 0.01 | -0.34 | -0.37 | 0.12 | 0.03 | 0.17 | 0.10 | 0.27 | -0.23 | -0.08 | 0.07 | -0.07 | -0.09 | 0.17 |
| $X_{16}$ | 0.00 | -0.09 | 0.31 | 0.11 | 0.16 | 0.16 | 0.04 | -0.07 | 0.49 | -0.21 | 0.08 | -0.18 | 0.08 | 0.23 | 0.42 | 0.14 | 0.06 | 0.11 | 0.09 | 0.19 | -0.24 | 0.04 |
| $X_{17}$ | -0.13 | 0.15 | -0.03 | 0.43 | -0.23 | -0.13 | -0.01 | 0.16 | -0.36 | 0.03 | -0.13 | -0.32 | 0.13 | 0.12 | 0.22 | -0.22 | 0.04 | 0.27 | -0.18 | 0.28 | 0.20 | -0.04 |
| $X_{18}$ | -0.32 | -0.15 | -0.18 | -0.05 | -0.10 | 0.32 | -0.06 | -0.07 | 0.15 | 0.17 | 0.12 | -0.03 | 0.36 | 0.26 | 0.08 | 0.43 | 0.05 | 0.06 | 0.08 | -0.02 | 0.40 | -0.04 |
| $X_{19}$ | 0.38 | -0.08 | -0.03 | -0.06 | 0.22 | 0.21 | -0.14 | -0.45 | 0.17 | -0.15 | 0.03 | 0.19 | -0.07 | -0.16 | 0.18 | -0.24 | -0.04 | 0.32 | -0.18 | -0.07 | 0.15 | -0.06 |
| $X_{20}$ | 0.47 | 0.14 | 0.16 | -0.16 | 0.34 | 0.30 | -0.01 | 0.06 | -0.21 | -0.15 | 0.01 | 0.04 | 0.34 | -0.19 | 0.00 | 0.11 | 0.03 | -0.05 | -0.15 | -0.01 | -0.12 | -0.22 |
| $X_{21}$ | 0.16 | 0.64 | -0.46 | -0.19 | -0.05 | -0.05 | 0.25 | -0.08 | 0.10 | -0.11 | 0.06 | 0.17 | 0.03 | 0.02 | 0.05 | 0.04 | -0.04 | 0.02 | -0.02 | 0.20 | -0.07 | 0.02 |
| $X_{22}$ | 0.17 | 0.62 | -0.44 | -0.13 | -0.05 | -0.01 | 0.32 | -0.11 | 0.14 | -0.15 | 0.03 | 0.13 | 0.01 | 0.01 | 0.00 | -0.04 | 0.08 | 0.06 | 0.10 | 0.10 | 0.08 | -0.02 |
| $X_{23}$ | 0.19 | 0.47 | -0.12 | 0.28 | -0.10 | -0.22 | 0.28 | 0.10 | 0.02 | 0.02 | 0.06 | -0.28 | 0.02 | 0.06 | 0.18 | 0.13 | 0.02 | 0.12 | -0.05 | -0.47 | -0.02 | 0.05 |
| $X_{24}$ | -0.02 | 0.16 | 0.09 | 0.21 | -0.05 | 0.47 | 0.13 | 0.13 | 0.10 | 0.20 | -0.12 | -0.02 | -0.54 | -0.07 | -0.18 | 0.31 | -0.08 | 0.13 | -0.24 | 0.15 | -0.02 | -0.01 |
| $X_{25}$ | 0.29 | 0.34 | 0.41 | 0.00 | -0.31 | 0.47 | 0.04 | 0.15 | 0.03 | 0.23 | 0.03 | -0.01 | 0.03 | -0.06 | 0.02 | 0.02 | -0.07 | 0.02 | -0.12 | -0.02 | 0.01 | 0.05 |
| $X_{26}$ | 0.16 | 0.10 | 0.19 | 0.08 | -0.32 | 0.09 | 0.07 | 0.05 | 0.32 | 0.48 | 0.07 | 0.13 | 0.23 | 0.17 | 0.01 | -0.39 | -0.01 | -0.28 | 0.05 | -0.01 | -0.05 | -0.14 |
| $X_{27}$ | 0.13 | 0.27 | 0.05 | 0.52 | -0.07 | -0.05 | -0.09 | 0.12 | -0.18 | -0.10 | 0.08 | 0.40 | 0.09 | -0.09 | 0.04 | 0.20 | 0.29 | -0.31 | -0.12 | -0.02 | -0.06 | 0.02 |
| $X_{28}$ | 0.05 | 0.11 | 0.01 | 0.09 | 0.22 | 0.04 | -0.47 | 0.46 | 0.17 | -0.13 | 0.02 | 0.31 | -0.02 | 0.27 | -0.04 | -0.21 | 0.07 | 0.10 | -0.13 | 0.06 | 0.05 | 0.13 |
| $X_{29}$ | 0.38 | -0.09 | -0.32 | 0.13 | 0.22 | -0.26 | -0.13 | 0.04 | -0.02 | 0.49 | -0.01 | 0.16 | 0.16 | -0.05 | 0.16 | 0.12 | -0.12 | 0.08 | -0.01 | 0.28 | -0.05 | 0.18 |
| $X_{30}$ | -0.13 | 0.35 | 0.39 | -0.33 | -0.05 | 0.21 | -0.05 | 0.09 | -0.12 | -0.24 | 0.06 | -0.20 | 0.08 | -0.13 | 0.05 | -0.10 | 0.28 | -0.10 | 0.14 | 0.18 | 0.08 | 0.39 |
| $X_{31}$ | 0.45 | -0.09 | -0.03 | 0.13 | 0.29 | -0.08 | -0.02 | 0.07 | 0.13 | 0.34 | -0.07 | -0.27 | -0.15 | -0.25 | 0.11 | 0.04 | 0.39 | 0.03 | 0.25 | -0.02 | -0.14 | 0.03 |
| $X_{32}$ | -0.24 | 0.03 | 0.28 | 0.01 | 0.06 | -0.09 | 0.18 | 0.48 | 0.14 | -0.06 | 0.27 | 0.30 | 0.08 | -0.05 | -0.11 | 0.05 | -0.05 | 0.40 | 0.26 | -0.15 | -0.04 | -0.06 |
| $X_{33}$ | 0.58 | -0.09 | 0.02 | 0.04 | 0.05 | 0.11 | -0.06 | 0.23 | -0.44 | -0.09 | 0.11 | -0.01 | 0.05 | 0.03 | 0.10 | 0.05 | -0.05 | 0.09 | 0.26 | 0.08 | 0.10 | -0.28 |

complete version can be found in the S1 Table.

$$F_1 = -0.381X_1 - 0.584X_2 - 0.032X_3 + \cdots + 0.447X_{31} - 0.237X_{32} + 0.583X_{33} \qquad (2)$$

$$F_2 = 0.385X_1 + 0.158X_2 + 0.195X_3 + \cdots - 0.086X_{31} + 0.033X_{32} - 0.093X_{33} \qquad (3)$$

$$F_3 = 0.307X_1 - 0.110X_2 + 0.431X_3 + \cdots - 0.031X_{31} + 0.257X_{32} - 0.019X_{33} \qquad (4)$$

$$\vdots$$

$$F_{21} = -0.012X_1 - 0.024X_2 + 0.102X_3 + \cdots - 0.142X_{31} - 0.043X_{32} + 0.097X_{33} \qquad (5)$$

**Table 6. Sample data of TSs.**

| No. of enterprise | $F_1$ | $F_2$ | $F_3$ | $F_4$ | $F_5$ | $F_6$ | $F_7$ | $F_8$ | $F_9$ | $F_{10}$ | $F_{11}$ | $F_{12}$ | $F_{13}$ | $F_{14}$ | $F_{15}$ | $F_{16}$ | $F_{17}$ | $F_{18}$ | $F_{19}$ | $F_{20}$ | $F_{21}$ | $F_{22}$ |
|---|---|---|---|---|---|---|---|---|---|---|---|---|---|---|---|---|---|---|---|---|---|---|
| 1 | 2.62 | 4.70 | 2.41 | 7.01 | 5.84 | 0.76 | 2.63 | 1.15 | 0.19 | -0.44 | -0.38 | 0.48 | -0.06 | 2.54 | 0.59 | 0.51 | 0.79 | -0.13 | 4.28 | 0.58 | 4.08 | -0.48 |
| 2 | 1.08 | 4.70 | 1.49 | 1.11 | 1.70 | -2.60 | 3.29 | -0.21 | 2.18 | 0.56 | 1.05 | -0.21 | 1.18 | 0.36 | 1.72 | 2.10 | 0.79 | 1.48 | 0.78 | 0.84 | 2.28 | 0.28 |
| 3 | 3.95 | 5.21 | -2.04 | 3.40 | 2.04 | -0.99 | -0.04 | 0.51 | 3.02 | -0.22 | 1.98 | -2.08 | 0.80 | 1.35 | -0.03 | 0.76 | -0.06 | 1.05 | 1.05 | 1.68 | 1.81 | 0.79 |
| 4 | 6.76 | 3.46 | -3.95 | 0.45 | 1.68 | 0.58 | 1.11 | 2.27 | -1.21 | 0.67 | 1.58 | -0.74 | -0.07 | 3.10 | 1.38 | 0.34 | 0.17 | 0.95 | 1.15 | 0.45 | 1.82 | 0.80 |
| 5 | 4.89 | -0.70 | 1.33 | 4.01 | 3.37 | -0.37 | 0.32 | 3.17 | 1.59 | -0.46 | 2.38 | -1.26 | 0.56 | 1.35 | 0.04 | 0.45 | -0.24 | 1.13 | 1.93 | 1.87 | 2.89 | -0.01 |
| 6 | 2.79 | 4.11 | 0.68 | -0.03 | 1.26 | -1.90 | 3.49 | 1.81 | 1.71 | -0.63 | 0.05 | 1.03 | 0.22 | 3.57 | 0.92 | 1.04 | 0.47 | 1.84 | 0.67 | 0.89 | 2.54 | 0.90 |
| 7 | 1.17 | 6.22 | -1.06 | 5.17 | 2.80 | -1.28 | 0.01 | -0.62 | 2.22 | 0.15 | 2.06 | 0.01 | 1.08 | 1.15 | 1.15 | 1.21 | 1.60 | 0.76 | 0.31 | 1.44 | 1.36 | 0.39 |
| 8 | 1.70 | 7.15 | 1.02 | 2.16 | 2.42 | -2.11 | 0.82 | 1.13 | 2.74 | -1.14 | 1.86 | -1.18 | 0.69 | 2.50 | -0.14 | 0.91 | -1.62 | 0.98 | 1.13 | 1.26 | 2.43 | -0.06 |
| 9 | 2.91 | 5.48 | -0.53 | 3.33 | 3.02 | -0.80 | 1.65 | 1.96 | 2.95 | -0.87 | 2.28 | -1.46 | 0.13 | 1.75 | 0.55 | 0.30 | -1.02 | 1.18 | 1.73 | 1.19 | 1.93 | 0.35 |
| 10 | 1.45 | 6.42 | 1.64 | 3.33 | 1.99 | -2.39 | 1.90 | 0.54 | 2.70 | -1.22 | 1.75 | 1.19 | 1.01 | 2.23 | 1.01 | 1.59 | 0.07 | 0.46 | 0.40 | 1.45 | 2.23 | -0.34 |
| ⋮ | ⋮ | ⋮ | ⋮ | ⋮ | ⋮ | ⋮ | ⋮ | ⋮ | ⋮ | ⋮ | ⋮ | ⋮ | ⋮ | ⋮ | ⋮ | ⋮ | ⋮ | ⋮ | ⋮ | ⋮ | ⋮ | ⋮ |
| 191 | 2.38 | 5.60 | -0.85 | 3.22 | 1.70 | -1.37 | -0.41 | 0.77 | 1.34 | -0.33 | 2.33 | 1.39 | 1.01 | 1.38 | -1.69 | 0.93 | 0.07 | 1.12 | 0.85 | 2.00 | 2.62 | 0.41 |
| 192 | 0.13 | 5.45 | -1.15 | 4.21 | 2.44 | -0.70 | 0.63 | 0.79 | 3.20 | -0.76 | 2.67 | -0.96 | 1.13 | 0.90 | 0.82 | 0.52 | 0.46 | 1.57 | 0.98 | 0.85 | 1.25 | 0.13 |
| 193 | -1.70 | 3.92 | 0.10 | 1.87 | 2.96 | -0.62 | 2.69 | 1.06 | 0.49 | -0.17 | 1.57 | 2.17 | 0.57 | 0.91 | 1.23 | -0.11 | -0.60 | 1.19 | 0.16 | 1.20 | 2.90 | 0.68 |
| 194 | -2.19 | 4.44 | 2.11 | 4.76 | 3.26 | -0.47 | -0.67 | 0.31 | 1.49 | -0.71 | 2.35 | 0.87 | 0.95 | 1.59 | 0.51 | 0.58 | 0.39 | 0.85 | 0.55 | 1.20 | 1.83 | -0.43 |
| 195 | 0.61 | 2.57 | 0.01 | 1.52 | 2.00 | -0.70 | 2.57 | 1.08 | 2.13 | -0.35 | 1.69 | 1.90 | 1.30 | 0.96 | 1.69 | 0.79 | -0.17 | 1.50 | 0.23 | 2.17 | 2.43 | 0.90 |
| 196 | -1.82 | 2.73 | 3.87 | 2.20 | 4.03 | -1.71 | -0.74 | 0.57 | 1.03 | 1.93 | 0.74 | 0.34 | 0.74 | 1.13 | 1.87 | 0.99 | -0.26 | 1.33 | 1.01 | 1.02 | 1.69 | 0.50 |
| 197 | 0.38 | 4.87 | -1.34 | 1.61 | 1.99 | -0.73 | 0.08 | 0.67 | 0.55 | 0.72 | 1.81 | 1.64 | 0.95 | 1.15 | -0.65 | 0.36 | -0.71 | 2.09 | 1.08 | 1.35 | 2.54 | 1.60 |
| 198 | -1.36 | -0.29 | 2.06 | 3.08 | 2.98 | -0.35 | 1.12 | 1.32 | 1.30 | 0.41 | 1.61 | 1.38 | 1.07 | 1.10 | 1.65 | 0.43 | -0.09 | 1.34 | 0.11 | 1.07 | 2.41 | 1.22 |
| 199 | -1.36 | 5.29 | -1.88 | 2.43 | 0.98 | -0.49 | 0.96 | 1.16 | 1.06 | -0.32 | 2.12 | 0.76 | 1.51 | 0.32 | 0.78 | 0.32 | -0.74 | 1.71 | 0.27 | 0.10 | 1.87 | 0.88 |
| 200 | 0.38 | 3.68 | -1.30 | 0.20 | 0.61 | -0.43 | 0.67 | 1.00 | 0.74 | 0.59 | 0.43 | 1.35 | 0.67 | 1.17 | 0.54 | 0.51 | 1.59 | 2.33 | 0.81 | 1.17 | 1.10 | -0.01 |

$$F_{22} = -0.150X_1 + 0.168X_2 - 0.033X_3 + \cdots + 0.029X_{31} - 0.055X_{32} - 0.282X_{33} \tag{6}$$

## 4. Prediction of TSs by BPNN

### 4.1 Three layers of BPNN

By reviewing the relevant studies on the evaluation index system, it can be found that most scholars used the AHP, Logistic regression, fuzzy evaluation method and other research methods to help to complete the process of establishing the evaluation system, and analyze the feasibility and accuracy. However, with the continuous development of artificial intelligence, artificial neural networks are used more and more widely in the field of evaluation and prediction, especially the BPNNM, applied in various industries. The advantage of BPNNM is that it can process lots of data and has objectivity. For instance, Sun & Lei [35] used BPNN to train and simulate the financial early warning model of listed mining companies, which proved the objectivity and practical value of this method.

Combined with the characteristics of BPNNM, and considering that the evaluation of TSs involves many and complex indicators, this paper finally selects BPNNM for evaluating TSs, and drew on the studies of related scholars. For example, Shu & Xu [36] combined multi-level dynamic fuzzy evaluation with BPNN, used multi-layer hidden layer neural network structure and BP algorithm to train the network, established an evaluation decision support system, and confirmed the feasibility of this model; Chen [37] used the AHP to calculate the index weights and used the BPNN method to simulate the samples, proving that the prediction accuracy was higher than that of the traditional prediction method.

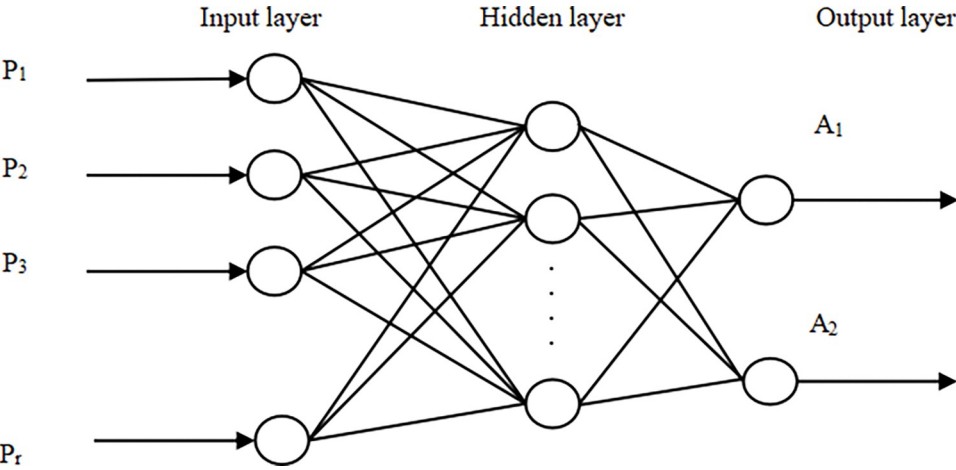

**Fig 1. BPNN topology.** It is a traditional three-layer BPNN structure.

BPNN is an artificial neural network obtained through a supervised learning (that is BP algorithm) [38]. Considering the problem of learning rate, the topology of BPNN generally has three layers, as shown in Fig 1, including input layer, hidden layer, and output layer. The output layer can be one or more [39].

The learning process of BPNN is mainly divided into two steps. The first step is to input the sample data and calculate the output value through the corresponding weight connection of each neuron between the input layer and the hidden layer. When the expected output is met, the process ends and the whole process is positive propagation. If the output layer does not get the expected output, it will enter the second step of BP, which is reverse propagation, send back the error signal, MSE, and correct the hidden layer to the output layer and the input layer to the hidden layer. Through repeated iterations, the process ends until the expected output value is obtained.

In this paper, three-layer BPNN is used, in which the input layer is the evaluation index. According to the above results of PCA, there are 22 principal components extracted. And there are 22 nodes in the input layer. The output layer is the result of evaluating TSs, that is, whether these start-ups are qualified, so the number of output layer node is 1. And MATLAB_R2017b software is used for programming to realize the training, simulation and evaluation for the model.

## 4.2 Confirmation of the number of nodes in the hidden layer

The setting of the number of nodes in the hidden layer of the BPNN is a critical part of this method, which has a great influence on the final results of calculation. There are 22 nodes in the input layer of BPNN, and there is 1 node in the output layer. According to the empirical formula, the number of nodes in the hidden layer ranges from 4 to 14. It is set that the maximum training times are 1,000, the required training precision is 0.1, and the learning rate is 0.1. The number of nodes in each hidden layer is simulated, and the results in the simulation are shown in Table 7.

It can be seen from Table 7 that when the number of nodes in the hidden layer is 10, the training accuracy is optimal and the number of iterations is minimum. Therefore, the number of nodes in the hidden layer in BPNN can be determined as 10.

$$y = \sqrt{z + l} + \beta \tag{7}$$

**Table 7. Network training efficiency of different hidden nodes.**

| No. of hidden layers | Iterations | Mean variance (MSE) |
|---|---|---|
| 4 | 10 | 0.09135 |
| 5 | 15 | 0.09471 |
| 6 | 7 | 0.09039 |
| 7 | 9 | 0.09037 |
| 8 | 9 | 0.08714 |
| 9 | 8 | 0.06686 |
| 10 | 6 | 0.05551 |
| 11 | 10 | 0.07726 |
| 12 | 8 | 0.08574 |
| 13 | 8 | 0.08828 |
| 14 | 7 | 0.09575 |

where, $y$ is the number of nodes in the hidden layer, $z$ is the number of nodes in the input layer, $l$ is the number of nodes in the output layer, and the value of $\beta$ is 1,2. . . 9,10.

### 4.3 The evaluation model based on BPNN

The evaluation model based on BPNN is established in this paper, and there is a three-layer neural network structure with 22 input layers, 11 hidden layers, and 1 output layer, as shown in Fig 2. The output of the BPNNM includes two classifications for evaluating TSs (qualified or unqualified enterprises). The calculation allows to evaluate the correct rate of investment. In turn, after simulation, the correct judgment rate of investing in qualified enterprises can be derived. In the simulation process, the data of 140 enterprises are randomly selected as training samples, among which, 70 enterprises are qualified and 70 enterprises are unqualified, and the data for the remaining 60 enterprises are selected as test samples, among which, 30 enterprises are qualified and 30 enterprises are unqualified.

### 4.4 Simulation results

After 10 times of simulation, the correct rate of evaluation based on BPNN is shown in Table 8.

It can be found that the correct rate of judging that the enterprises are qualified is between 23.33% and 90.00%, and the average correct rate is 58.33%. The correct rate of judging that the enterprises are unqualified ranges from 63.33% to 90.00%, and the average correct is 81.00%. It can be concluded that the results of evaluating the qualified enterprises are unstable. Although

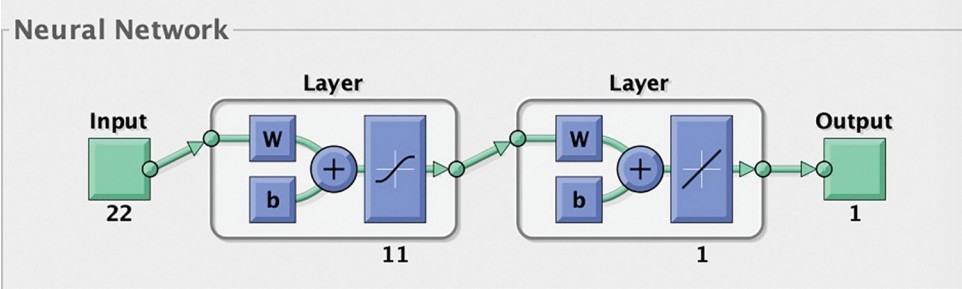

**Fig 2. Schematic diagram of the BPNN topology.** By specifying the number of nodes in the input layer, hidden layer and output layer and selecting the transfer function, the hierarchical structure of the network is determined.

**Table 8. Simulation results of the BPNNM.**

| Enterprise category | Simulation times | | | | | | | | | | Average correct rate (%) |
|---|---|---|---|---|---|---|---|---|---|---|---|
| | 1 | 2 | 3 | 4 | 5 | 6 | 7 | 8 | 9 | 10 | |
| Qualified enterprises | 90.00 | 46.67 | 23.33 | 76.67 | 76.67 | 60.00 | 30.00 | 60.00 | 66.67 | 53.33 | 58.33 |
| Unqualified enterprises | 86.67 | 86.67 | 63.33 | 76.67 | 90.00 | 86.67 | 83.33 | 83.33 | 86.67 | 66.67 | 81.00 |

most of the evaluation results are greater than 60.00% and the optimal accuracy rate of evaluation can reach 90.00%, there are still results with assessment accuracy of only 23.33%, 30.00% or 46.67%. However, the accuracy rate of evaluating unqualified enterprises is mostly more than 80.00%, but there are also cases where the evaluation accuracy rate is only 63.33%. It shows that BPNN can evaluate TSs based on sample data and has certain evaluation ability. Whereas, due to the shortcomings of BPNN itself, especially from the results of evaluating qualified enterprises. It can be clearly seen that the evaluation results based on BPNNM are not stable, and the average accuracy of evaluating the two types of enterprises is not very ideal. The stability of evaluation results is low, it is necessary to further optimize the model.

## 5. Prediction of TSs by GA-BPNNM

### 5.1 Combination of GA and BPNN

**5.1.1 Advantages of GA.** In order to make up for the shortcomings of the BPNNM, and improve the stability and accuracy of the simulation results, many scholars utilize the GA, gray wolf algorithm, particle swarm optimization algorithm to optimize the BPNNM, combining with the advantages of other methods to improve the performance of the neural network model. GA is an optimization algorithm inspired by biology, developing by referring to the theory of evolution and genetics [40], and following the survival of the fittest mechanism [41]. The whole process is a process of seeking a global optimal solution [42]. GA is an adaptive search method, which simulates several natural processes: selection, information, heredity, random mutation, and population dynamics [43,44]. GA is simple to calculate, using the encoding set of parameters rather than for the parameters themselves, and the optimization range is a point group rather than a single point, which has the advantages of good optimization effect and high optimization efficiency [45]. The general GA mainly has three operators: selection, crossover and mutation operator, which are also the core of GA. The basic execution process is as follows: initialization is carried out to generate initial population; then the individual evaluation is carried out to calculate the fitness of every individual and enter the link of population evolution [46]. According to the results of individual evaluation, some individuals are generally selected as mothers by roulette-wheel selection method, and chromosomes are respectively selected to cross according to the crossover probability $Pc$ to generate offspring. The new offspring mutate the chromosome according to the mutation probability $Pm$ to obtain a new generation of population. If the results meet the requirements, the calculation would be stopped; if not, the population evolution process would be carried out again. Experiencing continuous evolution, the optimal solution is obtained.

**5.1.2 Building GA-BPNNM.** It is found that GA has been applied to the related studies of neural network [47,48]. In particular, some scholars have pointed out that GA and BPNNM can complement the advantages with each other, which has high feasibility and accuracy [4,49]. Ding et al. proposed an optimization method with BP algorithm based on GA to speed up the training speed of BP algorithm and overcome the disadvantage that BP algorithm is easy to fall into the local minimum [50]. Using the University of California Irvine (UCI) machine learning repository data sets for experimental analysis, the results show that it is not

easy to fall into a local minimum to use the method of combining GA and BP to train neural network and has better generalization ability and stable performance. Yin et al. [45] proposed IGABP algorithm based on improved GA for public opinion prediction on Weibo platform (https://weibo.com/), a Chinese social media platform based on user relationships. The results proved that IGABP algorithm is feasible in public opinion prediction on Weibo platform and has better generalization ability and higher accuracy. Han et al. made use of the advantages of GA in optimization to optimize BPNN and build a GA-BP hybrid model, to effectively evaluate and screen scientific design schemes, which confirms that the GA-BP hybrid product modeling and designing evaluation system can quickly, conveniently, effectively, and scientifically evaluate design schemes [51]. Based on the advantages of GA in optimization, we try to select GA to optimize the BPNNM, establishing a GA-BPNN model (GA-BPNNM) and conduct relevant analysis.

GA-BPNNM is mainly based on both the advantages of BPNN such as fast search, and the advantages of GA to reach the optimal solution through global search to overcome the disadvantages of BPNN which is easy to fall into the local optimal. The combination of GA and BPNN can significantly improve the performance of neural network. The basic process is that the optimal solution of GA is decomposed as the initial weights and thresholds of BPNN, and the evaluation results of TSs are finally obtained through the learning and training process of BPNN. The construction of GA-BPNNM for evaluating TSs can reduce the subjective influence and improve the accuracy of the evaluation. The flow chart of the algorithm for the GA-BPNN is shown in Fig 3.

## 5.2 Parameter setting of GA-BPNNM and simulation

In the optimization process of BPNN by GA, selection, crossover and mutation operation are experienced. The model parameters are set as follows: the evolution algebra of gen is 100, the crossover probability of Pc is 0.6, and the mutation probability of Pm is 0.05.

In order to verify the optimization effect of GA on the BPNNM for the evaluation of TSs, the training sample and test sample methods are selected as shown above, and the optimized and unoptimized results are directly output.

As shown in Table 9, in the comparison of 10 simulation results, the average correct rate of evaluation for qualified enterprises is 80.33%, and the one of unqualified enterprises is 93.67%, and the average evaluation accuracy rate is increased by 22.00% and 12.67%, respectively. Even in the third simulation, when the evaluation accuracy rate of qualified enterprises in the BPNN is only 23.33%, the evaluation accuracy can still be improved to 70.00% after it is through GA optimization, and the average modeling time of GA-BPNNM is very little different from the average modeling time of BPNN. Therefore, GA can improve the accuracy of the evaluation results based on GA to a certain extent, but also make the stability of the evaluation results better, which confirms the feasibility of applying GA to optimize BPNN.

## 6. Conclusion

In summary, the following conclusions can be drawn:

1. This paper takes TSs as the research object, obtains the evaluation index system for the preliminary evaluation and screening of TSs. In the case of reducing the data information as little as possible, the dimension of the index is reduced, and the PCA method is used to extract the original 33 indicators into 22 unrelated principal components, which improves the speed and accuracy of the later analysis.

2. Through learning and simulating the BPNNM of the final data, from the simulation results for ten times, the highest evaluation accuracy rate of qualified enterprises can be 90.00%,

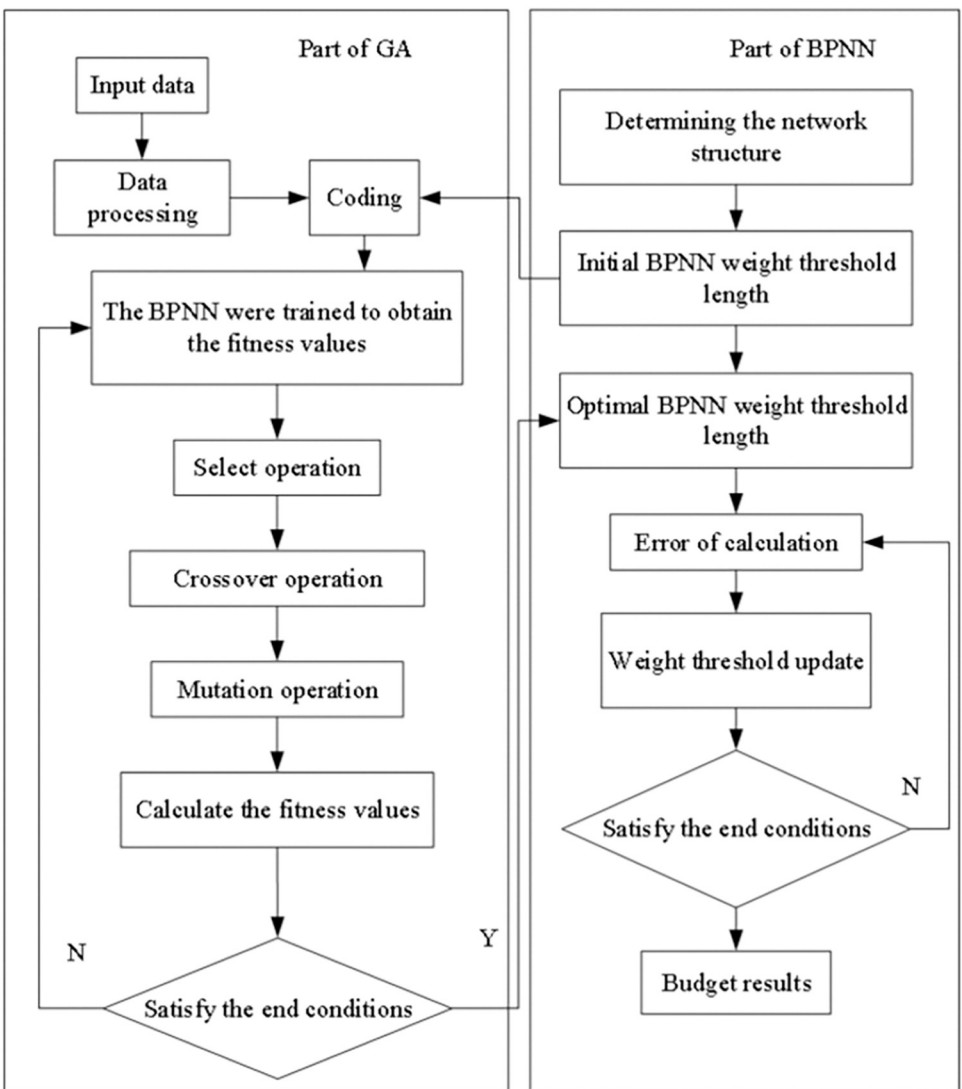

**Fig 3. Flow chart of the GA-BPNN algorithm.** The work chart of BPNN model based on genetic algorithm optimization was constructed.

the lowest is 23.33%, the range is 66.67%, and the average evaluation accuracy rate is 58.33%. The range of the evaluation accuracy rate of unqualified enterprises is 26.67%, and the average evaluation accuracy rate is 81.00%. The results obtained are not ideal, and there are problems with the unstable and low evaluation accuracy. The reason for this problem is that the weights and thresholds of the BPNNM are randomly generated, and the gradient descent algorithm is adopted, which leads to the result easily falling into the deficiency of the local optimal BP algorithm, so it is necessary to add related algorithms on this basis to optimize it and complement advantages with each other.

3. In view of the shortcomings of BP algorithm, this paper adopts GA to optimize it, uses the advantages of GA for global optimization, to take its optimal solution as the initial weights and thresholds of BPNNM, and obtains the GA-BPNNM. Empirical analysis is used to compare the simulation results of BPNNM before and after the improvement. From the simulation results for ten times, it can be seen, the range of the latter's evaluation accuracy

**Table 9. Simulation results of GA-BPNN and BPNN.**

| Test times | Accuracy rate of BPNN | | | Accuracy rate of GA-BPNN | | |
|---|---|---|---|---|---|---|
| | Modeling time (s) | Qualified enterprises (%) | Unqualified enterprises (%) | Modeling time (s) | Qualified enterprises (%) | Unqualified enterprises (%) |
| 1 | 1.90 | 90.00 | 86.67 | 0.86 | 70.00 | 93.33 |
| 2 | 0.71 | 46.67 | 86.67 | 1.01 | 86.67 | 100.00 |
| 3 | 0.58 | 23.33 | 63.33 | 0.57 | 70.00 | 86.67 |
| 4 | 0.63 | 76.67 | 76.67 | 0.62 | 86.67 | 90.00 |
| 5 | 0.60 | 76.67 | 90.00 | 0.56 | 83.33 | 96.67 |
| 6 | 0.58 | 60.00 | 86.67 | 0.60 | 90.00 | 93.33 |
| 7 | 0.59 | 30.00 | 83.33 | 0.57 | 80.00 | 93.33 |
| 8 | 0.65 | 60.00 | 83.33 | 1.10 | 76.67 | 96.67 |
| 9 | 0.52 | 66.67 | 86.67 | 0.55 | 80.00 | 93.33 |
| 10 | 0.59 | 53.33 | 66.67 | 0.59 | 80.00 | 93.33 |
| Average | 0.74 | 58.33 | 81.00 | 0.70 | 80.33 | 93.67 |

of the two types of enterprises is significantly lower than that of the former, the lowest evaluation accuracy of the latter is significantly higher than that of the former, the maximum optimization accuracy is 50.00%, and the average difference of modeling time is only 0.04 seconds. It shows that the BPNN optimized by GA significantly improves the evaluation accuracy, making the evaluation results more stable and reliable. The GA-BPNNM can be used for the evaluation and screening of TSs.

4. This paper attempts to use the artificial neural network model to input the data obtained from the evaluation system of TSs to get the simulation results, so that the evaluation results can be changed from subjective judgment to objective evaluation, and reduce the inevitable subjective factors in manual evaluation and screening. In addition, the GA-BPNNM established in this paper can be used to conduct preliminary evaluation of TSs. The relevant data of numerous enterprises can be input for simulation at one time, and the results can be output, which can greatly improve the efficiency and accuracy of preliminary screening work.

5. The evaluation system of TSs is a complex one that needs to be adjusted in time according to the changes of national policies and social development, whose influencing factors are numerous. In view of some requirements, including that the evaluation system ought to be scientific and rational, the one of TSs needs to be optimized and enriched in stages.

The experimental results show that the BPNNM optimized by the GA is superior to the BPNNM in both stability and accuracy of evaluation, and the modeling speed is basically the same. Meanwhile, the simulation results of this paper show that the GA-BPNNM can be used as an effective tool for the evaluation and screening of TSs, and providing an effective method of initially evaluating and screening TSs for various venture capital institutions. However, the number of samples selected in this paper is only 200, which is relatively small and has a certain impact on the evaluation results. Therefore, the number of samples should be expanded in the follow-up research to improve the accuracy of the research results.

## Supporting information

**S1 Table. The complete version of the sample data of TSs.**
(XLSX)

**S1 File. Data set.**
(ZIP)

## Author Contributions

**Methodology:** Jiaxin Li.

**Software:** Jiaxin Li, Yanjie Lv.

**Supervision:** Jian Yu.

**Writing – original draft:** Mingming Meng.

**Writing – review & editing:** Xin Liu, Jian Yu.

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
