## [Decision Letter · Decision Letter 0]

21 May 2023

PONE-D-23-13364Evaluation and Screening of Technology Start-ups Based on PCA and GA-BPNNPLOS ONE

Dear Dr. Yu,

Thank you for submitting your manuscript to PLOS ONE. After careful consideration, we feel that it has merit but does not fully meet PLOS ONE’s publication criteria as it currently stands. Therefore, we invite you to submit a revised version of the manuscript that addresses the points raised during the review process.

We look forward to receiving your revised manuscript.

Kind regards,

Lu Peng

Academic Editor

PLOS ONE

Journal Requirements:

   "This work was supported by the National Key Research and Development Program of the Ministry of Science and Technology of China (No. 2017YFB1402000). "

   "This work was supported by the National Key Research and Development Program of the Ministry of Science and Technology of China (No. 2017YFB1402000)."

   "This work was supported by the National Key Research and Development Program of the Ministry of Science and Technology of China (No. 2017YFB1402000). "

   "There is no any conflict of interests."

6. Please ensure that you include a title page within your main document. You should list all authors and all affiliations as per our author instructions and clearly indicate the corresponding author.

Reviewers' comments:

Reviewer's Responses to Questions

**Comments to the Author**

1. Is the manuscript technically sound, and do the data support the conclusions?

Reviewer #1: Yes

Reviewer #2: Partly

2. Has the statistical analysis been performed appropriately and rigorously? 

Reviewer #1: Yes

Reviewer #2: Yes

3. Have the authors made all data underlying the findings in their manuscript fully available?

Reviewer #1: Yes

Reviewer #2: Yes

4. Is the manuscript presented in an intelligible fashion and written in standard English?

Reviewer #1: Yes

Reviewer #2: No

5. Review Comments to the Author

Reviewer #1: This study has some significance and is written in a standard way, and I have only a few comments as follows:

1.How to choose the lag order of the inputs?

2.Review of the literature can be improved further, may include the study of other applications of BPNN model (https://doi.org/10.1016/j.dsm.2021.10.002;
https://doi.org/10.1016/j.energy.2022.123990).

3.Add several figures, such as method diagrams, result comparison diagrams, etc.

4.The formula is in image format, please change it.

5.Why choose GA to optimize BPNN instead of other evolutionary algorithms?

Reviewer #2: 1. Introduction can be improved. The scientific problem of this research can be described more clearly. Some real-world examples can be added to specify the research problem.

2. The clarity of the formula should be improved.

3. Literature review should be improved. Since this study adopt neural network and optimization algorithm. Researches about machine learning models and optimization techniques should be discussed. The following articles should be added.

[1]https://www.sciencedirect.com/science/article/abs/pii/S0020025523005091

[2]https://www.sciencedirect.com/science/article/abs/pii/S0020025522006260

[3]https://www.sciencedirect.com/science/article/pii/S2666764921000485

[4]https://www.sciencedirect.com/science/article/pii/S2666764922000364

[5]https://www.sciencedirect.com/science/article/abs/pii/S0957417422013057

[6]https://www.sciencedirect.com/science/article/abs/pii/S0957417421000129

4. The training result of BPNNM and GA-BPNNM can be presented and compared. For example, the Convergence curve when training the model.

5. The effectiveness of PCA should be evaluated by comparing the model without using PCA.

6. English quality should be improved.

6. PLOS authors have the option to publish the peer review history of their article (what does this mean?). If published, this will include your full peer review and any attached files.

Reviewer #1: No

Reviewer #2: No

---

## [Author Response · Author response to Decision Letter 0]

13 Jul 2023

Reviewer #1:

1. How to choose the lag order of the inputs?

Thank you for the suggestion. The BPNN model consists mainly of an input layer, an implicit layer and an output layer. The number of input and output layers is determined according to the actual problem, and a back-propagation algorithm of errors is used in learning and training, where the data is propagated layer by layer and the weights are continuously corrected so that the mean squared deviation of the output value from the true value meets the expectation. The number of neurons in the input layer of the model is equal to the number of input variables in the data to be processed, and the number of neurons in the output layer is equal to the number of outputs associated with each input. Instead of focusing on the lag order of the inputs, the model is initialized to determine the connections between the input, implicit and output layers in the initial state, the weight values, the threshold values for the implicit layer and the threshold values for the output layer.

2. Review of the literature can be improved further, may include the study of other applications of BPNN model 

 (https://doi.org/10.1016/j.dsm.2021.10.002;
https://doi.org/10.1016/j.energy.2022.123990).

Thank you for the suggestion. We have checked the literature carefully and added the above references into the literature review part in the revised manuscript. (Lines 19-20, page 4)

3. Add several figures, such as method diagrams, result comparison diagrams, etc.

Thank you for the suggestion. We have added the convergence curves of the GA-BPNN algorithm and the traditional BPNN (as shown in Figure 4).

4. The formula is in image format, please change it.

Thanks for your careful checks. We are sorry for our carelessness. Based on your comments, we have made the corrections. 

5. Why choose GA to optimize BPNN instead of other evolutionary algorithms?

The genetic algorithm has many advantages over other evolutionary algorithms: the GA is scalable and has a high degree of adaptability in combination with the BPNN model. At the same time, the search is inspired using evaluation functions and the process is simple. Moreover, the use of a probabilistic mechanism for iteration is stochastic and the search starts from a population, which is potentially parallel and allows for the simultaneous comparison of multiple individuals. In view of the shortcomings of BP algorithm, this paper adopts GA to optimize it, uses the advantages of GA for global optimization, to take its optimal solution as the initial weights and thresholds of BPNNM, and obtains the GA-BPNNM. 

Reviewer #2: 

1. Introduction can be improved. The scientific problem of this research can be described more clearly. Some real-world examples can be added to specify the research problem.

Thank you very much for your advice. This paper attempts to build an algorithm to reduce the cost of selection of investment targets for venture capitalists, in an attempt to solve the problem of adverse selection of startups that is common among venture capitalists in the investment process. We have amended the corresponding part of the text to make it clearer and more precise. (Lines 23-25, page 2)

2. The clarity of the formula should be improved.

Thank you for the suggestion. Modified throughout the paper according to the comment.

3. Literature review should be improved. Since this study adopt neural network and optimization algorithm. Researches about machine learning models and optimization techniques should be discussed. The following articles should be added.

[1]https://www.sciencedirect.com/science/article/abs/pii/S0020025523005091

[2]https://www.sciencedirect.com/science/article/abs/pii/S0020025522006260

 [3]https://www.sciencedirect.com/science/article/pii/S2666764921000485

 [4]https://www.sciencedirect.com/science/article/pii/S2666764922000364

[5]https://www.sciencedirect.com/science/article/abs/pii/S0957417422013057

[6]https://www.sciencedirect.com/science/article/abs/pii/S0957417421000129

We sincerely appreciate the valuable comments. We have checked the literature carefully and added the above references into the literature review part in the revised manuscript. (Lines 10-30, page 4)

4. The training result of BPNNM and GA-BPNNM can be presented and compared. For example, the Convergence curve when training the model.

Thank you for the suggestion. We have added the convergence curves of the GA-BPNN algorithm and the traditional BPNN (as shown in Figure 4, page 23).

5. The effectiveness of PCA should be evaluated by comparing the model without using PCA.

Thank you very much for your advice. We are sorry for the misunderstanding as we did not express it clearly. The use of PCA is necessary because the complexity and diversity of the influencing factors for start-ups requires the data related to all influencing factors to be downscaled. The use of PCA is necessary to filter and identify a total of 22 input variables, a necessary part of the determination of the number of input layers based on reality, to make the model more scientific.

6. English quality should be improved.

Thanks for your suggestion. We have invited a native English speaker from the USA to help polish our article. And we hope the revised manuscript could be acceptable for you.

According to the reviewer’s comments, we have revised the manuscript extensively. If there are any other modifications we could make, we would like very much to modify them and we really appreciate your help. We hope that our manuscript could be considered for publication in your journal. Thank you very much for your help.

---

## [Decision Letter · Decision Letter 1]

18 Oct 2023

PONE-D-23-13364R1Evaluation and Screening of Technology Start-ups Based on PCA and GA-BPNNPLOS ONE

Dear Dr. Yu,

Thank you for submitting your manuscript to PLOS ONE. After careful consideration, we feel that it has merit but does not fully meet PLOS ONE’s publication criteria as it currently stands. Therefore, we invite you to submit a revised version of the manuscript that addresses the points raised during the review process.

We look forward to receiving your revised manuscript.

Kind regards,

Mohamed Rafik N. Qureshi

Academic Editor

PLOS ONE

Additional Editor Comments:

The manuscript "Evaluation and Screening of Technology Start-ups Based on PCA and GA-BPNN" needs some more queries. Please refer to the detailed report.

Reviewers' comments:

Reviewer's Responses to Questions

**Comments to the Author**

1. If the authors have adequately addressed your comments raised in a previous round of review and you feel that this manuscript is now acceptable for publication, you may indicate that here to bypass the “Comments to the Author” section, enter your conflict of interest statement in the “Confidential to Editor” section, and submit your "Accept" recommendation.

Reviewer #1: All comments have been addressed

Reviewer #2: (No Response)

Reviewer #3: All comments have been addressed

2. Is the manuscript technically sound, and do the data support the conclusions?

Reviewer #1: Yes

Reviewer #2: (No Response)

Reviewer #3: No

3. Has the statistical analysis been performed appropriately and rigorously? 

Reviewer #1: Yes

Reviewer #2: (No Response)

Reviewer #3: No

4. Have the authors made all data underlying the findings in their manuscript fully available?

Reviewer #1: Yes

Reviewer #2: (No Response)

Reviewer #3: No

5. Is the manuscript presented in an intelligible fashion and written in standard English?

Reviewer #1: Yes

Reviewer #2: (No Response)

Reviewer #3: Yes

6. Review Comments to the Author

Reviewer #1: The author has revised my comments. Future revisions need to have the changes highlighted for easier review.

Reviewer #2: (No Response)

Reviewer #3: Paper title: Evaluation and Screening of Technology Start-ups Based on PCA and GA-BPNN

1) Please refer to "Table 1. (Technology) start-ups/ project assessment and evaluation index system '. Since based on a review of the literature the indicators are already finalized, "hence, Etc." may be removed under the first level indicators.

2) Please refer to "Using the Delphi method, through 6 experts and 9 investment practitioners and 6 rounds of anonymous discussion and communication, the final evaluation system and index score for TSs are determined, including 4 first-level indicators, 15 second-level indicators and 33 third-level indicators.“ the role of Delphi is crucial in the decision-making, How the selection of Delphi was carried out, their background and process of 6 rounds is crucial as it affects the outcome. Authors may provide more information on it.

3) Please refer to Table 2. Evaluation index system of TSs, various points are included in Table 2, the basis of allocation of these points and their use is unclear. The points may be written in a column adjacent to the "Code" column.

4) Please refer to "2.3 Data source ": what was the basis of "The selection criteria are as follows: "The data source was gathered through the questionnaire as mentioned in "One is through questionnaires that were filled out by these LTSs, and another is from information publicly disclosed by them." may be uploaded as a supplementary file along with collected data file.

5) The author may define qualified enterprises and unqualified enterprises also clarify the basis of classification, “Finally, data of 200 sample start-ups are obtained, and among these samples, 100 start-ups are qualified enterprises, and 100 unqualified."

6) The questionnaire pilot testing, questionnaire validity and response accuracy are important in an empirical investigation, which is missing.

7) Please refer to make the results obtained more scientific and accurate for this evaluation system, the PCA method is used for the final sample data to extract the principal components and eliminate the multicollinearity between indicators. The SPSS outcomes for factor loading in the manuscript may be provided.

8) There are some typos in the manuscript, for instance refer to the abstract: under purpose, the text is in mixed font sizes may be corrected.

7. PLOS authors have the option to publish the peer review history of their article (what does this mean?). If published, this will include your full peer review and any attached files.

Reviewer #1: No

Reviewer #2: No

Reviewer #3: No

---

## [Author Response · Author response to Decision Letter 1]

1 Dec 2023

On behalf of all the contributing authors, I would like to express our sincere appreciations of your letter and reviewers’ constructive comments concerning our article entitled “Evaluation and Screening of Technology Start-ups Based on PCA and GA-BPNN” (EMID: ac5e6676202ab79f). These comments are all valuable and helpful for improving our article. According to the reviewers’ comments, we have made extensive modifications to our manuscript to make our results convincing. In this revised version, changes to our manuscript were all highlighted within the document by using yellow-colored text. Point-by-point responses to the nice reviewers are listed below this letter.

Reviewer #1: The author has revised my comments. Future revisions need to have the changes highlighted for easier review.

We sincerely appreciate the valuable comments. We’ve highlighted the changes in yellow to make it easier to review. 

Reviewer #3: 

1. Please refer to “Table 1. (Technology) start-ups/ project assessment and evaluation index system”. Since based on a review of the literature the indicators are already finalized, “hence, Etc.” may be removed under the first level indicators.

Thank you very much for your advice. We have amended the corresponding part of the first level indicators to make it clearer and more precise. (Table 1, page 6)

2. Please refer to “Using the Delphi method, through 6 experts and 9 investment practitioners and 6 rounds of anonymous discussion and communication, the final evaluation system and index score for TSs are determined, including 4 first-level indicators, 15 second-level indicators and 33 third-level indicators.” The role of Delphi is crucial in the decision-making, how the selection of Delphi was carried out, their background and process of 6 rounds is crucial as it affects the outcome. Authors may provide more information on it.

Thank you for your valuable comments on the construction of the indicator system. We fully agree with the reviewer’s suggestions for modifications and acknowledge the lack of description of the implementation process of the Delphi method. We have added detailed implementation processes of the Delphi method to the manuscript, including the background characteristics of experts in the Delphi method and the specific implementation process. The newly added sections in the manuscript are as follows:

We utilized the Delphi method to further determine the indicators and their weights. Specifically, we selected six professors from universities and nine managers from related companies. All respondents were over 30 years old. Five professors from universities were all doctoral supervisors, and their research directions were all business management. The nine managers had more than five years of experience in technology investment companies or technology platform companies. Through online meetings, we solicited opinions from the aforementioned experts, distributed survey outlines and background resources in advance, and answered some experts’ queries. We summarized the experts’ first-round discussion opinions on the indicator system and the results to the experts for the second revision. After the fourth round of anonymous discussion, the indicator system was finally determined, and the experts’ opinions were unified. In the subsequent fifth and sixth rounds of discussion, we asked the experts to provide importance evaluations for the three-level indicators of the determined indicator system, thus further determining the corresponding scores for each indicator. The final evaluation system and index scores for TSs include 4 first-level indicators, 15 second-level indicators, and 33 third-level indicators. As shown in Table 2.

We thank the reviewer for his or her feedback and are pleased to have had the opportunity to address the issues raised. We hope that our additional explanations and clarifications have addressed the concerns raised in a satisfactory manner.

3. Please refer to Table 2. Evaluation index system of TSs, various points are included in Table 2, the basis of allocation of these points and their use is unclear. The points may be written in a column adjacent to the “Code” column.

 Thank you for your interest in the indicator system. We need to explain the source of the scores in Table 2. The scores for each indicator in Table 2 were obtained through the Delphi method. We conducted six rounds of Delphi discussions, with the first four rounds determining the indicator system, including the three-level indicators. In the fifth and sixth rounds, we asked experts to provide importance evaluations for each indicator in the indicator system, that is, to determine the corresponding scores for the indicators. Through two rounds of iterative discussions, we finally determined the corresponding scores for the three-level indicators. We have also added the implementation process of the Delphi method and the source of the indicator scores to the manuscript. The newly added sections in the manuscript are shown below：In the subsequent fifth and sixth rounds of discussion, we asked the experts to provide importance evaluations for the three-level indicators of the determined indicator system, thus further determining the corresponding scores for each indicator. 

4. Please refer to “2.3 Data source”: what was the basis of “The selection criteria are as follows: “The data source was gathered through the questionnaire as mentioned in”. One is through questionnaires that were filled out by these LTSs, and another is from information publicly disclosed by them.” may be uploaded as a supplementary file along with collected data file. 

Thank you for your valuable comments. We apologize for any misunderstandings. After a collective discussion among the authors and a re-examination of the original data, we discovered that there may have been some misunderstanding regarding the term “questionnaire” in the original manuscript. It should be clarified that we used the Delphi method to determine the indicators and weights, following which we collected the data through public disclosure and submitted the relevant data sets and codes as supplementary materials. In reality, the Delphi method is vastly different from the conventional questionnaire survey. In conclusion, this study’s data sources do not include data from questionnaire surveys, and we have revised the content of the manuscript accordingly. I deeply apologize for any misunderstandings caused by the previous incorrect expression.

5. The author may define qualified enterprises and unqualified enterprises also clarify the basis of classification, “Finally, data of 200 sample start-ups are obtained, and among these samples, 100 start-ups are qualified enterprises, and 100 unqualified.”

Thank you very much for your advice. We are sorry for the misunderstanding as we did not express it clearly. We rechecked the data sources section of the manuscript and found that more discussion on qualified firms was missing in this section. Therefore, we add the classification criteria of qualified enterprises and unqualified enterprises in Section 2.3.

6. The questionnaire pilot testing, questionnaire validity and response accuracy are important in an empirical investigation, which is missing.

We thank you for your comments. Our study did not obtain data through questionnaire survey. Our goal was to conduct expert discussions using the Delphi method to determine the indicator weights. This is fundamentally different from the traditional approach of using questionnaire surveys to gather enterprise data for analysis, and this method of determining the weights of indicators is similar to the entropy weight method.

The Delphi method is a structured consulting technique that reflects the collective opinion of experts, and it does not require additional reliability or validity testing. We recognize that it may have been unclear in the manuscript how the Delphi method was implemented, which may have led to this misconception. Therefore, we have included more detailed information on expert selection, the decision-making process of the Delphi method, and so on, to supplement the manuscript.

7. Please refer to make the results obtained more scientific and accurate for this evaluation system, the PCA method is used for the final sample data to extract the principal components and eliminate the multicollinearity between indicators. The SPSS outcomes for factor loading in the manuscript may be provided.

We greatly appreciate your comments. Upon examination of the manuscript, we noticed that the presentation of the results obtained through the principal component analysis method was not comprehensive enough. Therefore, in section 3.2, we have included more detailed results of the principal component analysis, including formulas and factor load calculations.

8. There are some typos in the manuscript, for instance refer to the abstract: under purpose, the text is in mixed font sizes may be corrected.

Thanks for your careful checks. We are sorry for our carelessness. Based on your comments, we have checked the literature carefully and made the corrections.

---

## [Decision Letter · Decision Letter 2]

5 Dec 2023

PONE-D-23-13364R2Evaluation and Screening of Technology Start-ups Based on PCA and GA-BPNNPLOS ONE

Dear Dr. Yu,

Thank you for submitting your manuscript to PLOS ONE. After careful consideration, we feel that it has merit but does not fully meet PLOS ONE’s publication criteria as it currently stands. Therefore, we invite you to submit a revised version of the manuscript that addresses the points raised during the review process.

We look forward to receiving your revised manuscript.

Kind regards,

Mohamed Rafik N. Qureshi, Ph.D.

Academic Editor

PLOS ONE

Journal Requirements:

Additional Editor Comments :

The manuscript entitled "Evaluation and Screening of Technology Start-ups Based on PCA and GA-BPNN" may be revised for minor revision. Further, the necessary complete data files may be uploaded as per PLOS ONE rule.

Reviewers' comments:

Reviewer's Responses to Questions

**Comments to the Author**

1. If the authors have adequately addressed your comments raised in a previous round of review and you feel that this manuscript is now acceptable for publication, you may indicate that here to bypass the “Comments to the Author” section, enter your conflict of interest statement in the “Confidential to Editor” section, and submit your "Accept" recommendation.

Reviewer #3: (No Response)

2. Is the manuscript technically sound, and do the data support the conclusions?

Reviewer #3: Yes

3. Has the statistical analysis been performed appropriately and rigorously? 

Reviewer #3: Yes

4. Have the authors made all data underlying the findings in their manuscript fully available?

Reviewer #3: No

5. Is the manuscript presented in an intelligible fashion and written in standard English?

Reviewer #3: Yes

6. Review Comments to the Author

Reviewer #3: Thank you for revising the manuscript.

(a) Furthermore, except remark number 6, the most of the comments have been satisfactorily resolved. The explanation appears to be a satisfactory one; nevertheless, a Delphi process content validity index could provide value to the manuscript.

(b) Data provided in the Table 5. Principal component factor load matrix looks incomplete and Table 6. Sample data of TSs look incomplete.

(c) A complete data file may be uploaded.

7. PLOS authors have the option to publish the peer review history of their article (what does this mean?). If published, this will include your full peer review and any attached files.

Reviewer #3: No

---

## [Author Response · Author response to Decision Letter 2]

11 Dec 2023

On behalf of all the contributing authors, I would like to express our sincere appreciations of your letter and reviewers’ constructive comments concerning our article entitled “Evaluation and Screening of Technology Start-ups Based on PCA and GA-BPNN” (EMID: ac5e6676202ab79f). These comments are all valuable and helpful for improving our article. According to the reviewers’ comments, we have made extensive modifications to our manuscript to make our results convincing. In this revised version, changes to our manuscript were all highlighted within the document by using yellow-colored text. Point-by-point responses to the nice reviewer are listed below this letter.

Reviewer #3: (a) Furthermore, except remark number 6, the most of the comments have been satisfactorily resolved. The explanation appears to be a satisfactory one; nevertheless, a Delphi process content validity index could provide value to the manuscript.

Thank you for your valuable comments. Due to space constraints, we had not initially included an indicator of content validity for the Delphi process in the manuscript. We thank the reviewer for bringing this to our attention, as we recognize that adding this section would enhance the rigor and scientific soundness of the article while adhering to the consistently high standards expected by the PLOS ONE journals. We have accordingly included the corresponding validity test content. The newly added sections in the manuscript are as follows: Prior to initiating the formal research of this paper, specifically prior to distributing the survey questionnaire, eight experts were requested to evaluate the validity of the questionnaire's structure, content, and overall design. Based on the evaluation results from these eight experts (see Tables 2.2), they provided positive feedback on the questionnaire structure, content, and overall design. Therefore, the survey questionnaire designed in this paper has a high level of validity.

Table 2.2 Result of experts' evaluation (N=8) 

 Fairly reasonable Basically sound Ordinary Somewhat unreasonable Completely unreasonable

Structure Number of choices 3 5 0 0 0

 Percentage 38% 63% 0% 0% 0%

Content Number of choices 2 6 0 0 0

 Percentage 25% 75% 0% 0% 0%

Overall 

design Number of choices 2 6 0 0 0

 Percentage 25% 75% 0% 0% 0%

(b) Data provided in the Table 5. Principal component factor load matrix looks incomplete and Table 6. Sample data of TSs look incomplete.

Thank you very much for your suggestion. We have added the corresponding parts of the principal component coefficient loading matrix to make it clearer and more precise. Limited to space, we have provided table 6 in the annex of the manuscript.

(c) A complete data file may be uploaded.

Thank you very much for your advice. We neglected the integrity of the data in the past, which led to a loss of rigor in the manuscript. We have once again verified all the data and additional attachments, and have included complete data as per the commenter's recommendations.

---

## [Editor Report · Decision Letter 3]

2 Jan 2024

Evaluation and Screening of Technology Start-ups Based on PCA and GA-BPNN

PONE-D-23-13364R3

Dear Dr. Yu,

We’re pleased to inform you that your manuscript has been judged scientifically suitable for publication and will be formally accepted for publication once it meets all outstanding technical requirements.

Kind regards,

Mohamed Rafik N. Qureshi, Ph.D.

Academic Editor

PLOS ONE

Additional Editor Comments (optional):

Thanks for making the necessary changes to your manuscript entitled "Evaluation and Screening of Technology Start-ups Based on PCA and GA-BPNN." Also, thank you for sticking to the PLOS ONE policy for data availability.
---

## [Editor Report · Acceptance letter]

26 Jan 2024

PONE-D-23-13364R3 

PLOS ONE

Dear Dr. Yu, 

I'm pleased to inform you that your manuscript has been deemed suitable for publication in PLOS ONE. Congratulations! Your manuscript is now being handed over to our production team.

Kind regards, 

on behalf of

Prof.(Dr.) Mohamed Rafik N. Qureshi 

Academic Editor

PLOS ONE